# Effect of Strain Rate on Tensile Properties of Carbon Fiber-Reinforced Epoxy Laminates with Different Stacking Sequences and Ply Orientations

**DOI:** 10.3390/polym15122711

**Published:** 2023-06-17

**Authors:** Donglin Gao, Zuguo Bao, Weijian Han, Xianpeng Wang, Shiyao Huang, Li Huang, Qiuren Chen, Hailong Zhao, Yahong Xu

**Affiliations:** 1College of Materials Science and Engineering, Nanjing Tech University, Nanjing 210009, Chinaweijianhan@njtech.edu.cn (W.H.); hli@njtech.edu.cn (L.H.);; 2Yangtze Delta Region Institute of Advanced Materials, Suzhou 215133, China

**Keywords:** CFRP, strain rate, stacking sequence, ply orientation

## Abstract

In practical application situations, a carbon fiber-reinforced polymer (CFRP) is often subjected to complex dynamic loadings. The effect of the strain rate on mechanical properties is very important for the CFRP design and product development. In this work, static and dynamic tensile properties of CFRP with different stacking sequences and ply orientations were investigated. The results showed that the tensile strengths of CFRP laminates were sensitive to the strain rate, while Young’s modulus was independent of the strain rate. Moreover, the strain rate effect was related to the stacking sequences and ply orientations. The experimental results showed that the strain rate effects of the cross-ply laminates and quasi-isotropic-ply laminates were lower than that of the unidirectional-ply laminates. Finally, the failure modes of CFRP laminates were investigated. Failure morphology demonstrated that the differences in strain rate effects among cross-ply laminates, quasi-isotropic-ply laminates, and unidirectional-ply laminates were caused by the mismatch between the fiber and the matrix when the strain rate increased.

## 1. Introduction

CFRP has been widely used in various industries, such as the aviation, aerospace, automotive, and shipbuilding industries, due to their excellent properties, including light weight, high strength, and resistance of corrosion [1,2,3,4]. However, composite material is often subjected to complex dynamic loading in the application stage, especially in the event of collisions [5,6]. It has been reported that the dynamic mechanical properties of materials are different from the results measured from quasi-static testing [7]. Research on collision safety in automobiles and airplanes found that the effects of the strain rate on the mechanical properties of materials were necessary to achieve a good correlation between the numerical simulation and actual service situations. In addition, composite materials are considered anisotropic materials in industrial design. Neglecting the mechanical behavior differences caused by layer-up design and fiber orientation may lead to design insufficiency, which hinders the wide application of carbon fiber composite materials. Thus, it is essential to investigate the effect of the strain rate on the mechanical properties and failure mechanisms of CFRP with different stacking sequences and ply orientations.

Previous researchers conducted numerous studies on the dynamic mechanical properties of CFRP. Hou and Ruiz et al. [8] tested dynamic tensile mechanical properties on weave CFRP and found that properties dominated by the fibers were hardly strain rate-sensitive, while properties dominated by the matrix were strain rate-dependent, as the resin was tougher. Yuanxin Zhou et al. [9] carried out both quasi-static and dynamic tensile tests on the bundles of carbon fiber. The results showed that the strain rate effect on the mechanical properties was negligible. Xuejie Zhang et al. [10] tested the tensile strength and Young’s modulus of CFRP, claiming that the mechanical properties of CFRP increased when the strain rate changed. Rajat Kapoor et al. [11] studied the mechanical properties of fiber-reinforced composites under high strain rates and found that the peak stress of composites was sensitive to the strain rate, with a threefold increase in peak stress when the strain rate increased from 1370 s^−1^ to 6066 s^−1^. Wang Jianjun et al. [12] investigated the dynamic mechanical behavior of CFRP at different strain rates. The result showed that the tensile strength and modulus of the materials were affected by the strain rate, demonstrating a logarithmic relationship between tensile strength and the strain rate. Other than the properties of the fiber and matrix, the dynamic mechanical properties of CFRP materials were also affected by many factors, such as temperature, fiber direction, and stacking sequences. Gilat et al. [13] studied CFRP with different fiber orientations using an SHPB (split-Hopkinson pressure bar) and found that the tensile strength of CFRP was sensitive to the strain rate, and the sensitivity was related to the fiber direction. Gu Kou et al. [14] performed a succession of load direction and off-axis tests on CFRP. The results showed that Young’s modulus and the tensile strength of laminate composites were independent of the strain rate in the load direction, while tensile strength increased with the increasing strain rate in the off-axis. Hosur et al. [15] tested CFRP with different stacking sequences. The result showed that the strength of the composite materials improved significantly at dynamic loadings, and the strain rate effect was related to stacking sequences.

Based on previous studies, it is difficult to draw the conclusions for the strain rate effect on the mechanical properties of CFRP. In addition, in actual situations, composite materials are always exposed to complex loads, and it is difficult to obtain comprehensive material parameters through mechanical testing in a single loading direction. Therefore, it is essential to conduct systematic research on the strain rate effect of composite materials with different stacking sequences and ply orientations under different strain rates.

This paper aimed to investigate the strain rate effect on CFRP laminates with different ply orientations and stacking sequences. Specimens of unidirectional-ply laminates, cross-ply laminates, and quasi-isotropic-ply laminates were prepared for the tensile testing under different strain rates. The strain rate effects on the mechanical properties and failure modes of CFRP laminates with different stacking sequences and ply orientations were discussed.

## 2. Materials and Methods

### 2.1. Materials

The pre-impregnated materials (USN20000), supplied by Weihai Guangwei Composite Material Co., Ltd. (Weihai, China), were used as raw materials for specimen preparation. The T300 carbon fiber was employed as the reinforcing fiber for the epoxy matrix, with a nominal thickness of 0.2 mm. The mass fraction of carbon fiber was 40%. To investigate the effects of the stacking sequences on the dynamic mechanical properties of CFRP laminates, three types of stacking sequences were designed, including unidirectional-ply laminate [0]_8_, cross-ply laminate [0/90]_2s_, and quasi-isotropic-ply laminate [0°/+45°/−45°/90°]_s_, as shown in Figure 1.

CFRP laminate composites were prepared by compression molding. As shown in Figure 2, the stacked, pre-impregnated materials were placed into a molding machine and cured at 120 °C for 90 min. After cooling to room temperature, a plate of laminate composite was obtained. Specimens with different ply orientations were obtained by cutting the composite plate at 0°, 45°, and 90° using a carving machine.

Similar to SHPB, specimens for high-speed tensile tests should be designed to satisfy one-dimensional stress propagation and stress uniformity requirements. Figure 3 provides the designed geometry of the tensile specimen, showing a rectangular bar of 120 mm × 10 mm × 1 mm. High-strength aluminum alloy tabs with dimensions of 40 mm × 10 mm × 1 mm were bonded to both ends of the specimens to avoid failure inside the gripping region. Quasi-static tensile tests used the same specimens as the high-speed tensile tests to ensure the comparability of the mechanical properties.

### 2.2. Mechanical Property Test

In this study, the tensile tests comprised two sets of quasi-static tests and three sets of dynamic tests. Dynamic tensile tests were conducted at the strain rates of 1 s^−1^, 10 s^−1^, and 100 s^−1^, and quasi-static tensile tests were conducted at the strain rates of 10^−3^ s^−1^ and 10^−2^ s^−1^. At least three duplicate specimens were tested for each material, and the mean values were provided.

The quasi-static tensile tests were carried out in the Yangtze Delta Region Institute of Advanced Materials (Suzhou, China) using a universal testing machine (Instron 5982). The test temperature was 25 °C ± 3 °C. The tests were conducted under displacement control. The strain of the specimen was measured using an automatic contact extensometer. The gauge length was set to 15 mm. Young’s modulus was calculated using the following equation:(1)$$E=\frac{∆\sigma }{∆\varepsilon }$$
where *E* is Young’s modulus, ∆σ is the stress increment of the stress–strain curve between 0.001 and 0.003, and ∆ε is the strain increment of stress–strain curve between 0.001 and 0.003.

Dynamic tensile tests were conducted in the Yangtze Delta Region Institute of Advanced Materials (Suzhou, China) using a high-speed testing machine (Instron VHS 160/100-20). The strain of the composite was measured by the Digital Image Correlation (DIC) system with a gauge length of 15 mm. The calculation for Young’s modulus was consistent with the quasi-static experiment.

### 2.3. Dynamic Stress Equilibrium

It was essential to maintain dynamic stress equilibrium during the dynamic tests. Stress waves had sufficient time to travel back and forth inside the specimen in the quasi-static state, achieving stress equilibrium before failure occurred. However, in the dynamic tests, the stress waves did not have enough time to reach stress equilibrium. Generally, stress waves needed to travel back and forth with the specimen more than three times to achieve stress equilibrium.

The velocity of the elastic stress wave was calculated using the following equations [16]:(2)$$c=\sqrt{\frac{E}{\rho }}$$
where *c* denotes the velocity of the stress wave, *E* is Young’s modulus, and *ρ* is the density of the CFRP;
(3)$$t=\frac{2L}{c}$$
where *L* denotes the length of the specimen between two grips, and *t* denotes the time of one stress wave traveling a round trip inside the specimen; and
(4)$$n=\frac{T}{t}$$
where *T* is the fracture time of the specimen, and *n* is the number of cycles of the stress wave.

In this study, Young’s modulus of the unidirectional-ply laminate was 137 GPa in the quasi-static state, and the density of the CFRP was 1.53 g/cm^3^. According to Equation (2), the velocity of the elastic stress wave was calculated as 9487 m/s. Then, the round-trip time of a stress wave was calculated as 8.44 μs, according to Equation (3). The fracture time of the unidirectional carbon fiber composites at 100 s^−1^ was approximately 160 μs, and the number of cycles of the stress wave was calculated to be more than 20 times, according to Equation (4). Other types of specimens had longer fracture times, which ensured that the specimens had enough time to reach dynamic stress equilibrium.

## 3. Results and Discussion

### 3.1. Mechanical Property Test

The stress–strain curves of unidirectional-ply laminates are presented in Figure 4. For the given ply orientation, quasi-static and the dynamic stress–strain curves were plotted together for comparison. In Figure 4a, it can be observed that the 0° specimen displayed typical elastic stress–strain behavior before fracture. When the strain rate reached 1 s^−1^, the stress–strain curve oscillated slightly, and the 0° specimen remained elastic before fracture. When the strain rate reached 10 s^−1^, the 0° specimen was prone to small-scale splitting when subjected to the tensile load, resulting in the specimen being pulled out from the gripping region. In the test with a strain rate of 100 s^−1^, all 0° specimens exhibited similar phenomenon, causing the failure of testing. Thus, the test results at 10 s^−1^ and 100 s^−1^ were discarded. The tensile strength and Young’s modulus of the 0° specimen were hardly influenced by strain rates. Figure 4b displays the stress–strain curve of unidirectional-ply laminates in the 45° direction. The 45° specimen at all strain rates exhibited conventional strain-rate responses, including initial linear responses, yielding, and a gradual transition to non-linear responses. Different from the 0° specimen, it seemed that the tensile strength and failure strain of the 45° specimen were influenced by the strain rate. The stress–strain curve of unidirectional-ply laminates at 90° was also plotted in Figure 4c, and it retained a linear relationship until the specimen failed. Similar to the 45° specimen, the tensile strength of the 90° specimen was also strain rate-dependent.

To further analyze the strain rate effect on material properties, tensile strength and Young’s modulus of unidirectional-ply laminates are summarized in Figure 5. As shown in Figure 5a, the tensile strength and Young’s modulus of the 0° specimen were hardly influenced by strain rates when the strain rate was increased. For the 45° specimen in Figure 5b, the tensile strength increased by 36% when the strain rate increased from 10^−3^ s^−1^ to 100 s^−1^. However, the Young’s modulus remained almost constant for varied strain rates. Similarly, for the 90° specimen in Figure 5c, the tensile strength increased significantly by 114% when the strain rate increased from 10^−3^ s^−1^ to 100 s^−1^, while the Young’s modulus was independent from the strain rate.

Generally, tensile properties of unidirectional-ply laminates at 0° and 90° were determined by carbon fibers and the resin matrix, respectively. From the previous reports, it is still unclear whether the strain rate effect of CFRP was mainly induced by carbon fiber or the resin matrix [17]. Based on the above experiment results, it could be concluded that the mechanical properties of carbon fiber were not sensitive to the strain rate. Instead, the resin matrix was enhanced with the increasing strain rate. It is also worth noting that tensile properties of 45° specimens were much lower than that of 0° specimens, indicating that only a minority of the carbon fibers were acting as reinforcement. As a result, the sensitivity of the tensile properties of the 45° specimen to the strain rate was between that of the 0° and 90° specimens. Therefore, the strain rate effect of unidirectional-ply laminates was mainly driven by the matrix.

Currently, the mechanism of the strain rate effect on CFRP is proposed based on different explanations. Some researchers claim that the strength of CFRP is dependent upon the strain rate [13], and the fracture of carbon fibers is contingent upon the microcrack initiation. At high strain rates, cracks lack enough time to grow and tend to split into fragments, resulting in the enhanced strength. In this study, the fracture morphology of CFRP was also observed to investigate the failure mechanisms of composites and the underlying causes of the strain rate effect. Figure 6 provides the failure morphologies of unidirectional-ply laminates under quasi-static and dynamic tensile experiments. In Figure 6a, the fracture morphologies of the 0° specimens are presented. The fractures of the 0° specimens mainly occurred near the clamping area, where the stress concentration was localized. The dominant fracture mode was bundles of carbon fiber splitting. As the strain rate increased, the specimen split into more fragments. However, the tensile strength of the 0° specimen was proved to be independent of the strain rate. Figure 6b illustrates the fracture morphologies of the 45° specimens. The fractures of the specimens were primarily localized in the gauge length section. The failure morphologies of the specimens were relatively consistent, breaking along the 45° direction without significant delamination or fiber bundles splitting. The failure morphologies of the specimens were predominantly determined by the matrix and their interfacial bonding with fibers. In addition, the failure morphologies of 45° specimens remained consistent under different strain rates. Similarly, the fractures of 90° specimens were mainly localized in the gauge length section. The fracture direction was perpendicular to the load direction, with regular and flat cross-section surfaces. The specimen failure was driven by the matrix at all strain rates.

Previous tensile testing demonstrated that tensile strengths of both 45° and 90° specimens were sensitive to the strain rate. Meanwhile, the matrix and interfacial bonding were dominant factors of failure mode for these specimens. Thus, we could conclude that the main cause of the strain rate effects of unidirectional-ply laminates was related to matrix and interfacial bonding.

### 3.2. Cross-Ply Laminates

In cross-ply laminates, specimens at 0° and 90° possessed identical fiber directions. Therefore, only 0° and 45° specimens of cross-ply laminates were investigated in this section.

The stress–strain curves of cross-ply laminates are shown in Figure 7. In Figure 7a, obvious fluctuations in the stress–strain curve were observed under dynamic loadings, which were attributed to the vibration in the test equipment. Similar phenomena were also reported in previous studies [10]. Cross-ply laminates exhibited nearly elastic behavior before fracturing at the 0° direction, as shown in Figure 7b. For the 45° specimens of cross-ply laminates, the stress–strain curves were composed of both elastic and plastic stages after yield under all strain rates. The 45° specimen exhibited obvious “pseudo plasticity” characteristics [18]. The 45° specimen did not completely fail after reaching the yield point and still maintained a certain bearing capacity. In quasi-static tests, as the tensile test progressed, the matrix was gradually damaged. A part of the fiber lost the constraint of the matrix and reorientated under the shear load, providing enhanced strength against the tension. As a result, the tensile strength of 45° specimens increased to some extent after the yield point. At a strain rate of 100 s^−1^, failure of the matrix occurred instantaneously, indicating a brittle failure process, with neglectable fiber reorientation. Thus, the tensile strength was hardly enhanced after the yield point. When the strain rates were 1 s^−1^ and 10 s^−1^, the stress–strain curve of the material exhibited pseudo-plasticity, similar to the quasi-static state. The mechanical behaviors of the 45° specimens at strain rates of 1 s^−1^ and 10 s^−1^ were between 10^−3^ s^−1^ and 100 s^−1^.

In addition, the failure strain of the 45° specimen was much higher than that of the 0° specimen. Compared with the 45° specimen of unidirectional-ply laminates, the 45° specimen of cross-ply laminates possessed a higher tensile strength, Young’s modulus, and failure strain, indicating that properties of CFRP laminates were highly influenced by stacking sequences and ply orientation.

The tensile strength and Young’s modulus of the cross-ply laminate were plotted against the strain rate in Figure 8. It was observed that the tensile strength of cross-ply laminates at 0° and 45° increased with the increased strain rate. However, Young’s modulus was independent of the strain rate by fluctuating within a certain range. When the strain rate exceeded 10 s^−1^, the tensile strength of 0° specimens exhibited significant strain rate sensitivity, while the 45° specimen exhibited the opposite trend. As the ply orientation approached 0°, cross-ply laminates tended to be more sensitive to the strain rate. This was different from the unidirectional-ply laminates, where 0° specimens were independent of the strain rate. It indicated that the strain rate effect of CFRP was also influenced by stacking sequences and ply orientation.

The fracture morphologies of cross-ply laminates under varied strain rates are provided in Figure 9. The fracture of the 0° specimen, as shown in Figure 9a, was mainly located near the clamping point and perpendicular to the load direction. The cross-section displayed a relatively rough surface and remained unchanged with varied strain rates. Figure 9b illustrates the fracture morphology of the 45° specimen. Although the carbon fiber bundles in the 45° specimen appeared intact in the static test, in reality, the entire specimen completely failed and could be separated into pieces with little force. The fracture area was visible in the gauge length section with a V-shaped surface, resulting mainly from matrix damage and interface failure during the tension.

According to previous results, as shown in Figure 4, both 90° and 45°specimens of unidirectional-ply laminates possessed lower failure strains than 0° specimens. It demonstrated that when the matrix, interface, and fiber were simultaneously subjected to tensile loads, the failures of the matrix and the interface occurred before that of the carbon fiber. Therefore, from the lateral fracture morphology of the 0° specimen of cross-ply laminates in Figure 10, their failure mode was the mixture of delamination, matrix breakage, interface debonding, and fiber breakage. In a high strain rate situation, the matrix and interface were enhanced by the strain rate effect, leading to more obvious fiber breakage during the failure. For the 45° specimen of cross-ply laminates, the deformation response was dominated by shear, displaying the failure modes of matrix breakage and interface delamination. As shown in Figure 11, such a failure mode was not affected by the strain rate.

Compared with the 0° specimen of cross-ply laminates, the 45° specimen displayed a lower sensitivity to the strain rate. The reason was probably related to the failure mechanism of the composites. Based on previous analysis, the strain rate effect on CFRP laminates was mainly driven by resin matrix and interface bonding. The resin was enhanced as the strain rate increased, while the tensile strength of carbon fiber remained unchanged. It aggravated the mismatching between the fiber and the matrix during the deformation, facilitating the crack initiation and propagation in the specimen. For cross-ply laminates, cracks propagated not only in the fibers and matrix but also along the interlayer, causing delamination and cracking in the specimen. As a result, resin deformation was delayed to some extent, indicating less sensitivity to the strain rate for cross-ply laminates. Compared to the 0° specimen, delamination of the 45° specimen occurred at a lower strain rate, yielding a lower strain rate sensitivity of the tensile strength.

### 3.3. Quasi-Isotropic-Ply Laminates

For quasi-isotropic-ply laminates, only the properties of the 0° specimens were investigated, since specimens at 0°, 45°, and 90° possessed identical fiber directions. The stress–strain curves of quasi-isotropic-ply laminates under tensile loads are illustrated in Figure 12. The stress increased nearly linearly with the strain at the initial stage, and yielding occurred as the strain approached failure.

The tensile strength and Young’s modulus of quasi-isotropic-ply laminates were plotted against the strain rate in Figure 13. As the strain rate was elevated from 10^−3^ s^−1^ to 100 s^−1^, the tensile strength of the 0° specimen increased by 22%, while the Young’s modulus fluctuated in a wide range. Similar to cross-ply laminates, the strain rate sensitivity of the tensile strength in quasi-isotropic-ply laminates was lower than that of unidirectional-ply laminates.

Figure 14 displays the fracture morphologies of quasi-isotropic-ply laminates. The fracture of the specimen was primarily located in the gauge length section. The cross-section of the laminates appeared relatively rough, with evident interlayer cracking and interfacial debonding. The failure morphology exhibited little change with increasing strain rates.

Figure 15 displays the lateral fracture morphology of quasi-isotropic-ply laminates under varied strain rates. The rough cross-section and delamination fracture clearly indicated that the failure mode of the specimen was a mixture of delamination, matrix breakage, fiber breakage, and interface debonding. The length of the pull-out fibers increased as the strain rate increased. It was reported that longer fiber pull-out could extend the crack path and absorb more energy, leading to the increase in tensile strength [19].

## 4. Conclusions

This study investigated the strain rate effect on the tensile properties of CFRP laminates with different ply sequences, including unidirectional-ply laminates, cross-ply laminates, and quasi-isotropic laminates. In addition, the failure modes of CFRP laminates were analyzed and discussed. The main conclusions can be drawn as follows:(1)Generally, the tensile strengths of CFRP laminates were sensitive to the strain rate. The sensitivity was related to the stacking sequences and ply orientations, while Young’s modulus was independent of the strain rate.(2)The strain rate effect of CFRP laminates was mainly caused by the resin matrix and interface bonding. For unidirectional-ply laminates, the tensile strength of 90° specimens displayed strong strain rate effects. Additionally, the sensitivity of the tensile properties of the 45° specimen to the strain rate was between that of the 0° and 90° specimens.(3)The strain rate sensitivity of the tensile strengths of the cross- and quasi-isotropic-ply laminates were lower than that of unidirectional-ply laminates. The difference in the strain rate effect between the matrix and the fiber facilitated the crack initiation and propagation in the cross- and quasi-isotropic-ply laminates, delaying the resin matrix deformation, which was the main attributor for the strain rate effect of the composites.

## Figures and Tables

**Figure 1 polymers-15-02711-f001:**
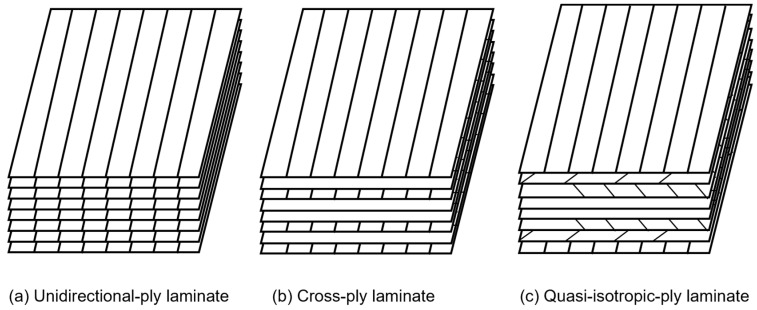
Stacking sequences.

**Figure 2 polymers-15-02711-f002:**
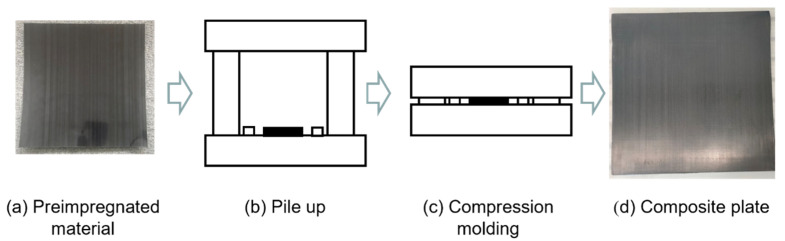
Composite material compression molding process.

**Figure 3 polymers-15-02711-f003:**
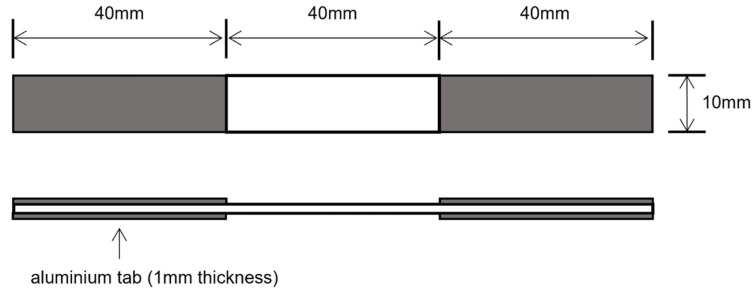
Dimensions of the testing specimen for tests.

**Figure 4 polymers-15-02711-f004:**
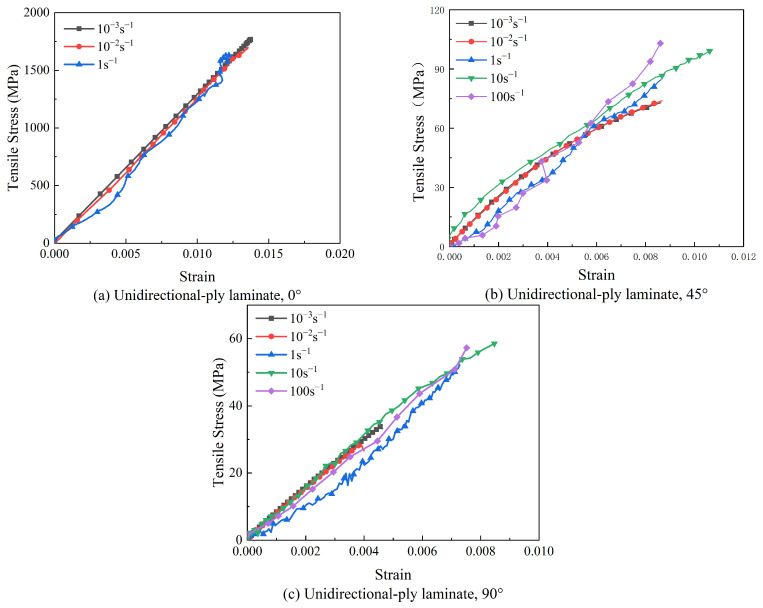
Stress–strain curves of unidirectional-ply laminates at different strain rates.

**Figure 5 polymers-15-02711-f005:**
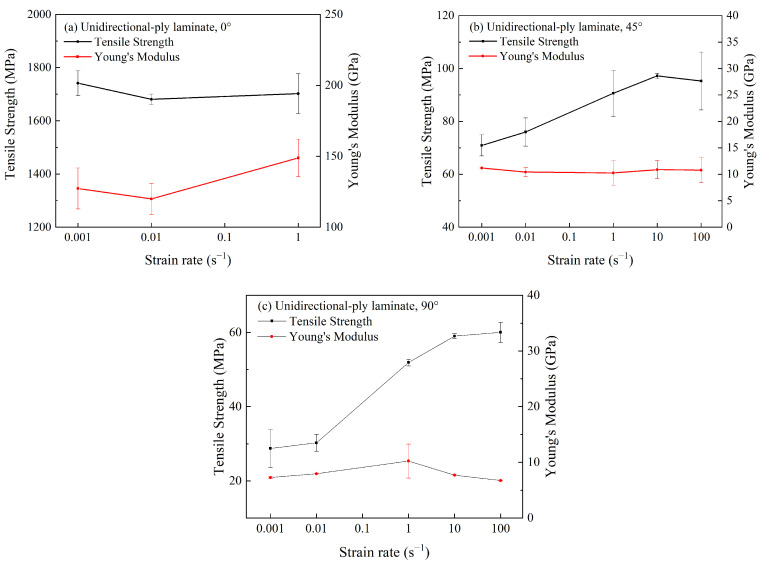
Tensile strength and Young’s modulus of unidirectional-ply laminates at different strain rates.

**Figure 6 polymers-15-02711-f006:**
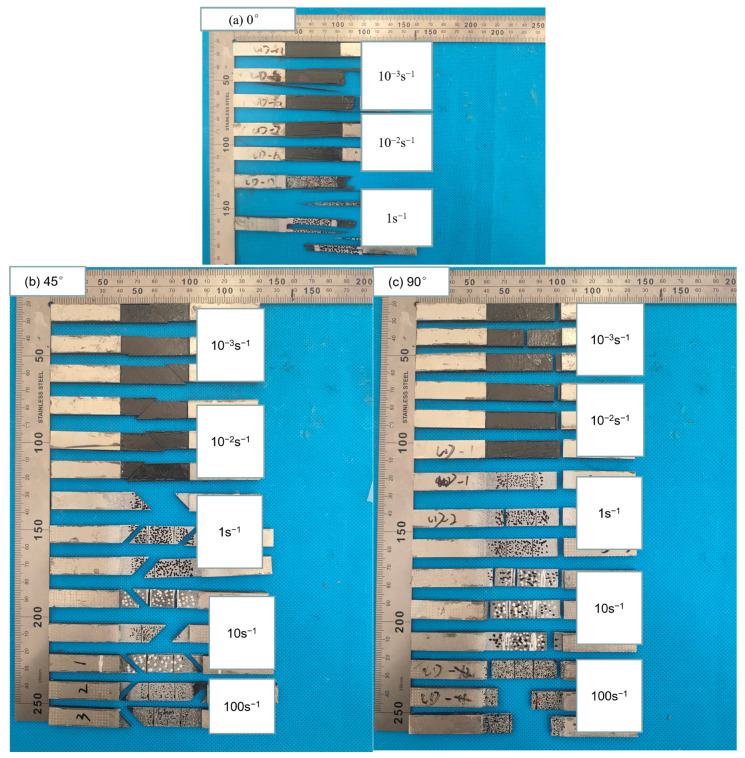
Failure morphologies of unidirectional-ply laminates at different strain rates.

**Figure 7 polymers-15-02711-f007:**
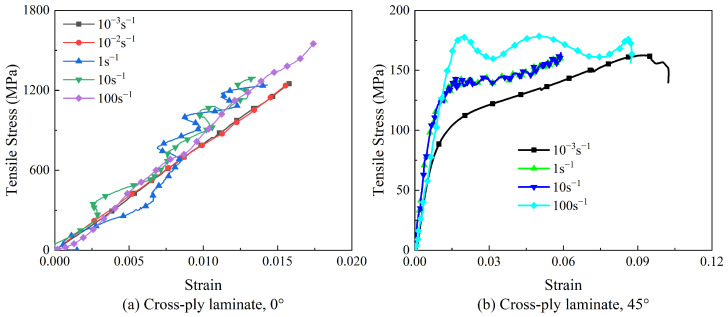
Stress–strain curves of cross-ply laminates at different strain rates.

**Figure 8 polymers-15-02711-f008:**
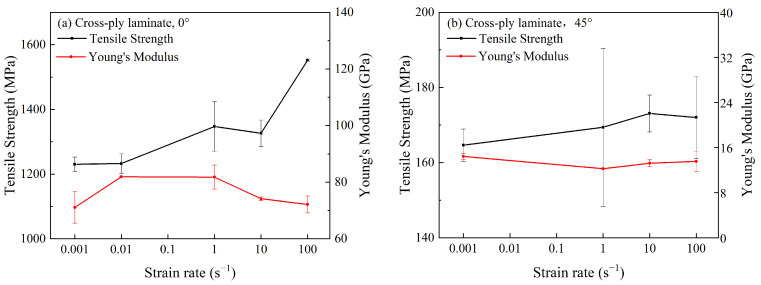
Tensile strengths and Young’s moduli of cross-ply laminates at different strain rates.

**Figure 9 polymers-15-02711-f009:**
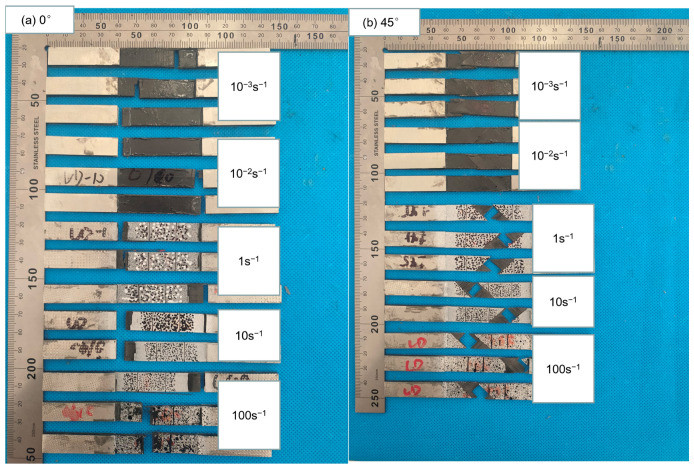
Failure morphologies of cross-ply laminates at different strain rates.

**Figure 10 polymers-15-02711-f010:**
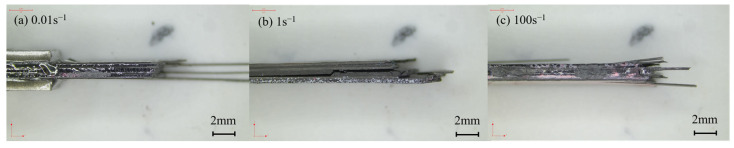
Failure morphologies of the sides of the 0° specimens of cross-ply laminates at different strain rates.

**Figure 11 polymers-15-02711-f011:**
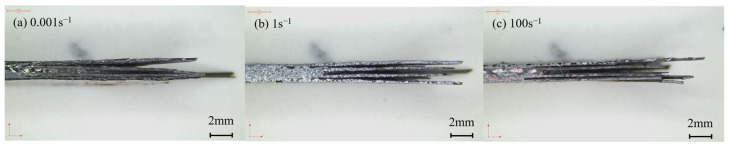
Failure morphologies of the sides of the 45° specimens of cross-ply laminates at different strain rates.

**Figure 12 polymers-15-02711-f012:**
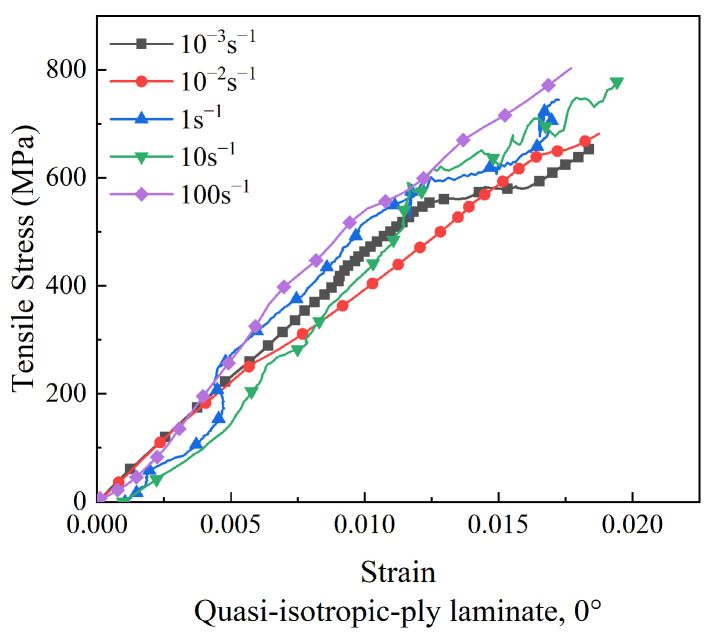
Stress–strain curves of quasi-isotropic-ply laminates at different strain rates.

**Figure 13 polymers-15-02711-f013:**
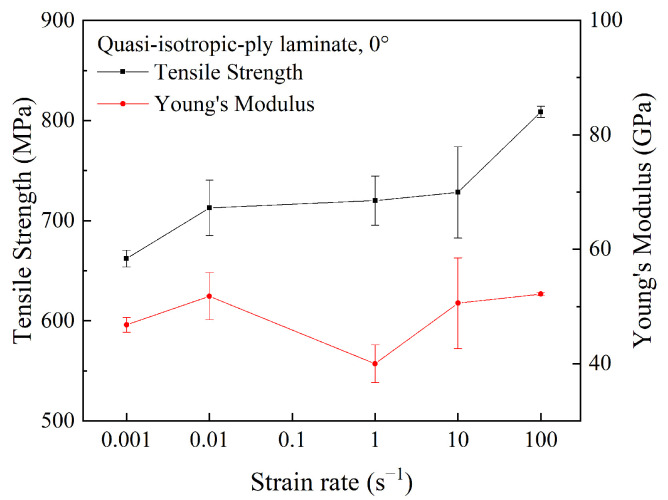
Tensile strength and Young’s modulus of quasi-isotropic-ply laminates at different strain rates.

**Figure 14 polymers-15-02711-f014:**
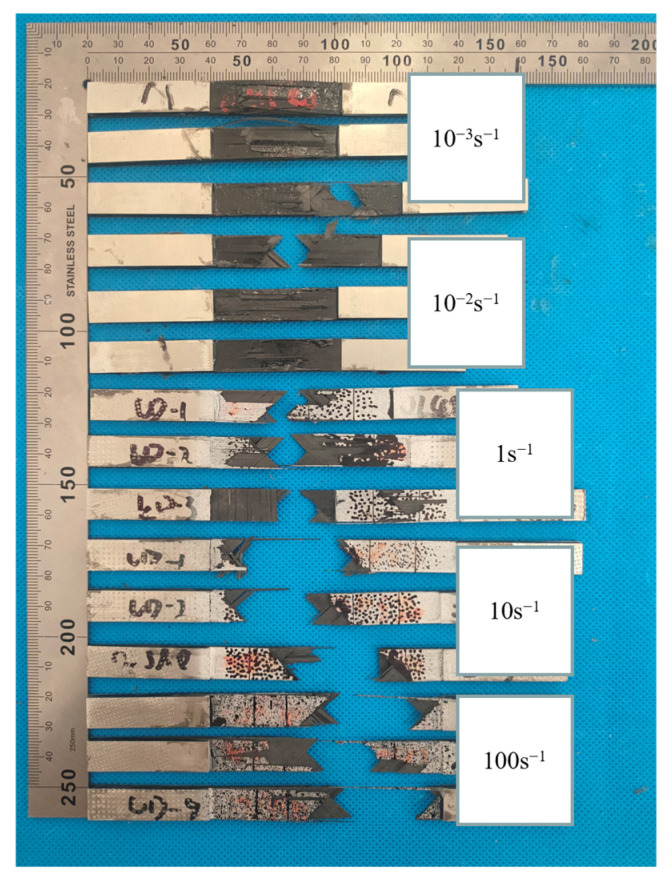
Failure morphologies of quasi-isotropic-ply laminates at different strain rates.

**Figure 15 polymers-15-02711-f015:**
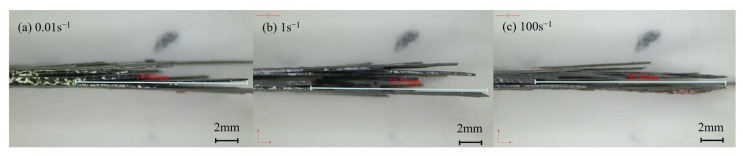
Failure morphologies of the sides the 0° specimens of quasi-isotropic-ply laminates at different strain rates.

## Data Availability

Data will be made available upon request.

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
