# Peer review of "Effect of Strain Rate on Tensile Properties of Carbon Fiber-Reinforced Epoxy Laminates with Different Stacking Sequences and Ply Orientations"

_polymers, 2023, doi:10.3390/polym15122711_

Round 1
Reviewer 1 Report
Gao et al. chose different carbon fiber reinforced polymers with different stacking consequences and ply orientations to investigate the effects of strain rate on mechanical properties. The authors designed the experiment well, but the work's importance is unclear. Please review the work carefully to describe why the work is important and the insight of the work. I suggested the major revision and resubmitting the paper.
Page 4, line equation 1, or notation of the C, please verify C is a capital letter or a lowercase.
Figure 4, what is the unit of Strain?
In general, how many tests for each sample or each factor? There is no error bar for tensile strength or Young’s modulus. Please add the error bar and average value if the experiment repeats several times.
Figure 4a shows no data about the performance under 10 S-1 and 100 S-1. If there are not many differences under the low frequency, the authors should add the data to conclude because other samples have the performance under these two strain rates. Please add related data.
The author discussed different orientations and the effects of the strain rate. What is the insight of this work? Why is this work important? Is any directions or applications that can conclude for fundamental research or applied engineering? If the orientation or strain rate is important, which conditions the author suggested for testing different materials?
Author Response
Dear reviewer,
Thank you for your letter about the reviewers’ comments on our manuscript titled “Effect of strain rate on tensile properties of carbon fiber reinforced epoxy laminates with different stacking sequences and ply orientations” (Polymers-2430883). These opinions help to improve academic. Based on your suggestion and request, we have made corrected modifications on the revised manuscript. We appreciate for reviewers’ warm work earnestly, and hope that the correction will meet with approval. Once again, thank you very much for your comments and suggestions.
We prepared a detailed responses to comments, which was listed as the appendix of this letter.
Thanks for your consideration.
Sincerely yours,
Zuguo Bao, Ph.D
College of Materials Science and Engineering
Nanjing Tech University, Nanjing, China
E-mail: baozuguo@njtech.edu.cn
Responses to the reviewers’ comments
- Q: Page 4, line equation 1, or notation of the C, please verify C is a capital letter or a lowercase.
A: Thanks for your comment. The letter c is lowercase. I make modifications in the revised manuscript.
- Q: Figure 4, what is the unit of Strain?
A: Thanks for your comment. Strain refers to the relative deformation of an object locally under external forces and other factors. ’Strain is calculated using the following equation:
(1) |
Where is the strain, ∆L is the length of the gauge length segment after deformation, and L is the length of the gauge length segment before deformation.
Strain is dimensionless physical quantity and is usually expressed as a percentage.
- Q: In general, how many tests for each sample or each factor? There is no error bar for tensile strength or Young’s modulus. Please add the error bar and average value if the experiment repeats several times.
A: Thanks for your comment. Each sample should be tested at least three times. The error bar has been added to the revised manuscript. It is worth noting that in Figure 7(b), under the strain rates of 1s-1, the discretization of tensile strength data of 45° specimens of orthogonal laminates is very serious, which is related to its failure mode. The shear failure of the 45° specimen exhibits obvious "pseudo plasticity" characteristics, and the CFRP do not completely fail after reaching the yield point and still maintains a certain bearing capacity. In quasi-static tests, as the tensile test progresses, the matrix gradually fails. Fiber that loses the constraint of the matrix will deflect under shear load, and some fiber will fail due to reaching their tensile strength ahead of time. Another fiber that does not reach their tensile strength continue to bear the load. Therefore, after the yield point, the tensile strength of the material continues to increase. At 100s-1, failure occurs simultaneously, and the failure process is almost brittle. When the strain rates are 1s-1, the strain-strain curve of the material exhibits pseudo plasticity similar to quasi-static state. However, with the strain rate increasing, the deformation after yield point decreased which make the stress-strain curve change after the yield point more complex that leads to the serious discretization of tensile strength data of 45° specimens of orthogonal laminates. Although the tensile strength had a certain degree of dispersion, the effect of strain rate on the tensile strength was significant. Similar phenomena have been observed in other literature.
- Q: Figure 4a shows no data about the performance under 10S-1 and 100S-1. If there are not many differences under the low frequency, the authors should add the data to conclude because other samples have the performance under these two strain rates. Please add related data.
A: Thanks for your comment. Figure 4(a) shows the tensile strength and Young's modulus of a single layer laminated plate 0° specimen at different strain rates. At strain rates of 10-3s-1 and 10-2s-1, the stress-strain curve of the 0° specimen remains elastic before fracture. When the strain rate reaches 1s-1, the stress-strain curve oscillates, but the 0° specimen remains elastic before fracture. When the strain rate is 10s-1, the 0° specimen is prone to small-scale splitting when subjected to tensile load, resulting in the specimen being pulled out of the gasket. In the test with a strain rate of 100s-1, all 0° specimens exhibited this phenomenon, so the test results at 10s-1 and 100s-1 strain rates are discarded. In Even if comparing the mechanical properties of unidirectional ply laminates at strain rates of 10-3s-1-1s-1. The 45° and 90° specimen exhibit significant strain rate effects, while the tensile strength of the 0° specimen is independent of strain rate.
- Q: The author discussed different orientations and the effects of the strain rate. What is the insight of this work? Why is this work important? Is any directions or applications that can conclude for fundamental research or applied engineering? If the orientation or strain rate is important, which conditions the author suggested for testing different materials?
A: Thanks for your comment. This article studied strain rate effect of CFRP laminates with different stacking sequences and ply orientations. The experimental results showed that in the strain rate range of 10-2s-1 to 100s-1, the strain rate effect of the 90° unidirectional laminates was the most significant, with an increase in tensile strength of 114%, while the 0° specimens showed almost no growth; The tensile strength of both cross ply laminates and quasi-isotropic ply laminates increased with the increase of strain rate, and the increase in tensile strength of the specimens ranges from 5% to 28%, which was much lower than that of the 90° specimens of unidirectional ply laminates. The strain effect of unidirectional ply laminates was not only related to the mechanical properties of the fiber and resin matrix itself, but also to the stacking order.
CFRP is widely used in the fields of automobiles and aerospace due to their lightweight, high-strength characteristics, and flexible design capabilities. However, previous research on collision safety in automobiles and airplanes have found that using material parameters obtained in quasi-static state to study the dynamic mechanical properties of materials will yield incorrect results. The material parameters measured in high-speed tensile tests can be used, then a good correlation between numerical simulation and real situations can be achieved. In addition, CFRP are considered anisotropic materials in industrial design. Neglecting the mechanical behavior changes caused by changes in layer order may lead to insufficient design, which hindering the wider application of CFRP. Therefore, it is very important to study the dynamic mechanical behavior of carbon fiber composite materials with different layering sequences and directions. The significance and application scenarios of this study has been added in the relevant section of the Introduction.
In addition to carbon fiber composite materials, SMC is widely used in the automotive industry due to its excellent performance and flexible processability. However, there is a lack of research on the dynamic mechanical behavior of SMC, neglecting the mechanical behavior changes caused by dynamic loading, which may result in insufficient design and hinder its wider application. Therefore, it is very important to study the effect of strain rate on the mechanical properties of SMC.

Reviewer 2 Report
The manuscript entitled "Effect of strain rate on tensile properties of carbon fiber rein-forced epoxy laminates with different stacking sequences and ply orientations" deals with the evaluation of the influence of the strain rate on the mechanical properties and failure mechanisms of carbon-reinforced composites.
in my view, before recommending the manuscript for publication, the following issues have to be solved:
- In paragraph 2.1, please check the sentence "A nominal thickness of 0.2 mm".
- Please, specify the ISO/ASTM standard followed for the mechanical tests.
- Please, add the standard deviation values in Figures 5, 8, 13.
- The Authors should specify the method used to evaluate the Elastic modulus.
- In paragraph 3.1, the Authors stated that "The tensile strength and Young's modulus of the 0° specimen were hardly influenced by strain rates". Below, it was reported that: "Different from the 0° specimen, it seemed that the tensile strength and failure strain of the 45° specimen was influenced by the strain rate.". Please verify and explain.
- Stress-strain curves reported in Figure 7a (as well as those depicted in Figure 12) appear very irregular. Please, explain the reason and clarify how the Young's modulus was evaluated.
The English language needs some minor revisions, please check the whole manuscript in order to correct all typos.
Author Response
Dear reviewer,
Thank you for your letter about the reviewers’ comments on our manuscript titled “Effect of strain rate on tensile properties of carbon fiber reinforced epoxy laminates with different stacking sequences and ply orientations” (Polymers-2430883). These opinions help to improve academic. Based on your suggestion and request, we have made corrected modifications on the revised manuscript. We appreciate for reviewers’ warm work earnestly, and hope that the correction will meet with approval. Once again, thank you very much for your comments and suggestions.
We prepared a detailed responses to comments, which was listed as the appendix of this letter.
Thanks for your consideration.
Sincerely yours,
Zuguo Bao, Ph.D
College of Materials Science and Engineering
Nanjing Tech University, Nanjing, China
E-mail: baozuguo@njtech.edu.cn
Responses to the reviewers’ comments
- Q: In paragraph 2.1, please check the sentence "A nominal thickness of 0.2 mm".
A: Thanks for your comment. The sentence has been corrected as “T300 carbon fiber was employed as reinforcing fiber for the epoxy matrix, with a nominal thickness of 0.2 mm.”
- Q: Please, specify the ISO/ASTM standard followed for the mechanical tests.
A: Thanks for your comment. There are no national or international standards that can be referenced for high-speed tensile testing of continuous fiber reinforced plastic. The method of dynamic tensile testing is still under development. The current research results indicate that the quantitative criteria for effective SHPB testing are also applicable to dynamic tensile testing. In order to obtain effective stress-strain data in material testing, the specimen should be in a stress equilibrium state, SHPB research has determined that in order to achieve dynamic stress balance, the loading pulse should propagate back and forth more than three times within the sample. The detailed calculation method is presented in paragraph 2.3.
- Q: Please, add the standard deviation values in Figures 5, 8, 13.
A: Thanks for your comment. The standard deviation values has been added to Figures 5, 8, 13. It is worth noting that in Figures 7 (b), under the strain rates of 1s-1, the discretization of tensile strength data of 45° specimens of orthogonal laminates is very serious, which is related to its failure mode. The shear failure of the 45° specimen exhibits obvious "pseudo plasticity" characteristics, and the CFRP do not completely fail after reaching the yield point and still maintains a certain bearing capacity. In quasi-static tests, as the tensile test progresses, the matrix gradually fails. Fiber that loses the constraint of the matrix will deflect under shear load, and some fiber will fail due to reaching their tensile strength ahead of time. Another fiber that does not reach their tensile strength continue to bear the load. Therefore, after the yield point, the tensile strength of the material continues to increase. At 100s-1, failure occurs simultaneously, and the failure process is almost brittle. When the strain rates are 1s-1, the strain-strain curve of the material exhibits pseudo plasticity similar to quasi-static state. However, with the strain rate increasing, the deformation after yield point decreased which make the stress-strain curve change after the yield point more complex that leads to the serious discretization of tensile strength data of 45° specimens of orthogonal laminates. Although the tensile strength has a certain degree of dispersion, the effect of strain rate on the tensile strength is significant. Similar phenomena have been observed in other literature.
- Q: The Authors should specify the method used to evaluate the Elastic modulus.
A: Thanks for your comment. We had specified the calculation for the Elastic modulus of composites as followed. “Young's modulus is calculated using the following equation:
(1) |
Where E is Young's modulus, and is the stress increment of stress-strain curve between 0.1% and 0.3% and is the increment of stress-strain curve between 0.1% and 0.3%.”
- Q: In paragraph 3.1, the Authors stated that "The tensile strength and Young's modulus of the 0° specimen were hardly influenced by strain rates". Below, it was reported that: "Different from the 0° specimen, it seemed that the tensile strength and failure strain of the 45° specimen was influenced by the strain rate.". Please verify and explain.
A: Thanks for your comment. In Figure 4(a), the tensile strength of the 0° specimen is very close, and Young's modulus is also the same. Thus, we can conclude that "The tensile strength and Young's modulus of the 0° specimen were hardly influenced by strain rates".
In Figure 4(b), at high strain rate (1s-1, 10s-1, and 100s-1), the tensile strength of the 45° specimen is significantly higher than that at low strain rate (10-3s-1, and 10-2s-1). Thus, we concluded that “the tensile strength and failure strain of the 45° specimen was influenced by the strain rate.”
- Q: Stress-strain curves reported in Figure 7a (as well as those depicted in Figure 12) appear very irregular. Please, explain the reason and clarify how the Young's modulus was evaluated.
A: Thanks for your comment. This is related to the system ringing. The pulses during the load introduction process can cause oscillation in the testing system, which is called system ringing. The system ringing can cause non-uniform deformation of the specimen. Despite the occurrence of system ringing, the 45° specimen of unidirectional orthogonal laminates still exhibits significant linearity before fracture.
The Young's modulus of CFRP is determined using the secant modulus between 0.1% and 0.3% strain. After the system ringing phenomenon occurs at a strain of 0.26%, the impact on the Young's modulus is relatively small.

Round 2
Reviewer 1 Report
Thanks for the author’s response. The manuscript can be accepted before the authors make the minor changes:
I understand the strain is typically expressed to a percentage. In your experiment part, you describe the calculation of Young’s modulus with the strain between 0.1% to 0.3%. But in your figures 4,7, and 12, the strain was not expressed as a percentage. It is the dimensionless ratio. Please keep the consistency of the units. Otherwise, the reader is confused about which part you used for Young’s modulus calculation. You can choose to convert the strain in figures to percentages or rewrite the description in your experiment to dimensionless value.
Author Response
Dear reviewer,
Thank you for your letter about the reviewers’ comments on our manuscript titled “Effect of strain rate on tensile properties of carbon fiber reinforced epoxy laminates with different stacking sequences and ply orientations” (Polymers-2430883). These comment and suggestions help to improve academic. Based on your suggestion and request, we have made corrected modifications on the revised manuscript. We appreciate for reviewers’ warm work earnestly, and hope that the correction will meet with approval. Once again, thank you very much for your comments and suggestions.
Thanks for your consideration.
Sincerely yours,
Zuguo Bao, Ph.D
College of Materials Science and Engineering
Nanjing Tech University, Nanjing, China
E-mail: baozuguo@njtech.edu.cn
Responses to the reviewers’ comments
Comments and Suggestions for Authors 1#:
I understand the strain is typically expressed to a percentage. In your experiment part, you describe the calculation of Young’s modulus with the strain between 0.1% to 0.3%. But in your figures 4,7, and 12, the strain was not expressed as a percentage. It is the dimensionless ratio. Please keep the consistency of the units. Otherwise, the reader is confused about which part you used for Young’s modulus calculation. You can choose to convert the strain in figures to percentages or rewrite the description in your experiment to dimensionless value.
Answer: Thanks for your comment and suggestion. I convert the strain in my manuscript to dimensionless value. In paragraph 2.2, the sentence has been corrected as “Where E is Young's modulus, and ∆σ is the stress increment of stress-strain curve between 0.001 and 0.003 and ∆ε is the strain increment of stress-strain curve between 0.001 and 0.003.”

Reviewer 2 Report
The manuscript can be accepted for publication as the Authors modified the text following the Reviewer's suggestions
Author Response
Thanks for your encouraging comment.